# Multi-Layer Defences for Robust GNSS Timing Retrieval [note 1]

**DOI:** 10.3390/s21237787

**Published:** 2021-11-23

**Authors:** Ciro Gioia, Daniele Borio

**Affiliations:** European Commission Joint Research Centre, Via Enrico Fermi, 2749, 21027 Ispra, Italy; daniele.borio@ec.europa.eu

**Keywords:** interference defences, RIM, T-RAIM, jamming

## Abstract

A multi-layered interference mitigation approach can significantly improve the performance of Global Navigation Satellite System (GNSS) receivers in the presence of jamming. In this work, three levels of defence are considered including: pre-correlation interference mitigation techniques, post-correlation measurement screening and Fault Detection and Exclusion (FDE) at the Position, Velocity, and Time (PVT) level. The performance and interaction of these receiver defences are analysed with specific focus on Robust Interference Mitigation (RIM), measurement screening through Lock Indicators (LIs) and Receiver Autonomous Integrity Monitoring (RAIM). The case of timing receivers with a known user position and using Galileo signals from different frequencies has been studied with Time-Receiver Autonomous Integrity Monitoring (T-RAIM) based on the Backward-Forward method. From the experimental analysis it emerges that RIM improves the quality of the measurements reducing the number of exclusions performed by T-RAIM. Effective measurements screening is also fundamental to obtain unbiased timing solutions: in this respect T-RAIM can provide the required level of reliability.

## 1. Introduction

Time and frequency synchronisation directly impact many market segments. For example, they are crucial for accurate and reliable time-stamping (recording of date and time) in financial transactions and are employed by telecommunication infrastructures and for energy network monitoring. In some cases, the device used for time and frequency synchronisation is installed in remote areas with difficult access to network infrastructures such as fibre optics. In such conditions, a solution can be the use of Global Navigation Satellite System (GNSS) timing receivers, which are easy to operate and have low infrastructural requirements.

GNSS is considered an important timing source, which is free, easy to access and highly accurate [1]. The 2019 Galileo Market report [2] discusses the use of GNSS timing in critical infrastructures such as telecom, energy and finance. The key requirements for such applications are also discussed and, in particular, the stakeholder consultation showed that robustness is considered fundamental by the users. The need of increased robustness is justified by the progressive spread of GNSS threats such as jamming.

Even if illegal, cheap jamming devices are still available in popular e-commerce platforms and an increased number of interference events has been reported in several studies [3,4]. For instance, the STRIKE3 project, in its final report [5], described a large number of interference events originating from a wide variety of interference and jamming signals. Thus, a robust GNSS device should take into account different approaches to detect and mitigate interference, which can assume different forms. In the timing context, the development of robust GNSS technologies has been promoted by the European Commission, which has founded projects such as GalilEo Authenticated Robust timing System (GEARS) and Galileo-based tIming receiver for increasing criticAl iNfrastructures rObustness (GIANO) [6] to develop robust Galileo timing receivers with specific focus on critical infrastructures.

A GNSS receiver for timing retrieval is made of several functional blocks, from the antenna to the final timing solution computation. Thus, several layers of defence can be implemented along the whole receiver processing chain to improve performance in the presence of interference [7]. Multiple defences are also included in the standardization action proposed by the European Commission [8].

In this work, three layers of defence are considered: at the pre-correlation level exploiting Robust Interference Mitigation (RIM) [9], post-correlation level measurement screening based on Lock Indicator (LIs) [10] and finally at the navigation solution level using Receiver Autonomous Integrity Monitoring (RAIM)/Fault Detection and Exclusion (FDE) [11].

While a multi-layer approach can significantly improve receiver performance, little analysis has been conducted on the interactions between the different layers of defence. The impact of the mitigation techniques has been assessed in [9] in the measurement and position domains, whereas the impact of five pre-correlation techniques was evaluated considering timing receivers in [12]. In these cases, LIs and Time-Receiver Autonomous Integrity Monitoring (T-RAIM) were not analysed.

The interaction among the different layers of interference mitigation has only been marginally investigated in the literature. Thus, the main contribution of this work is to experimentally evaluate the performance and interaction of these receiver defences with specific focus on RIM and T-RAIM. This paper complements the results presented in [13]. With respect to the conference paper, the analysis focuses on Galileo timing: only Global Positioning System (GPS) was considered in [13]. Additional metrics such as the maximum of the residuals and Timing Protection Level (TPL) have been analysed and an additional experiment using GNSS signals on the E5B band and with a different modulation has been considered.

The experimental analysis is based on the setup and the data described in [9].

From the analysis it emerges that pre-correlation techniques enhance the quality of the measurements and reduce the number of exclusions performed by the integrity algorithm. The adoption of multiple layers of defence against interference led to a more continuous and reliable timing solution. In some cases, the receiver was able to provide a reliable timing solution even when the receiver front-end was highly saturated by the interference signal.

The remainder of the paper is organised as follows: the different interference defences are described in Section 2. Section 3 summarises the experimental setup developed for the tests, while experimental results are analysed in Section 4. Finally, conclusions are drawn in Section 5.

## 2. Interference Defences

A schematic representation of the different processing blocks implemented for the evaluation of timing solutions is shown in Figure 1: interference defence blocks are indicated in light azure: (1) pre-correlation interference mitigation techniques, (2) Phase Lock Indicator (PLI) and (3) T-RAIM.

The PLI can be considered a post-correlation defence whereas T-RAIM performs measurement screening at the navigation/timing solution level.

These three layers of defence are described in the following sections.

### 2.1. Pre-Correlation Defences

Pre-correlation interference mitigation techniques operate directly on the samples provided by the receiver front-end. A new stream of samples is produced where the impact of interference should be reduced. The four RIM techniques described in [9] are considered in this work. A general RIM technique operates by first translating a set of *N* samples into a transformed domain where interference admits a sparse representation [14]. Failure in obtaining a sparse interference representation may lead to significant performance degradation.

In the transformed domain, a non-linearity is applied to reduce the impact of the samples affected by interference. Finally, the samples are brought back into the time domain. Two domains, time and frequency, are considered here along with two non-linearities. In this way, four RIM techniques are obtained. In particular, the following techniques have been analysed:Time Domain Pulse Blanking (TDPB): also known a Pulse Blanking (PB). This technique sets to zeros all the input samples with magnitude greater than a decision threshold, $T_{h}$. In this case, $T_{h}=3\sigma$ was selected where $\sigma^{2}$ is the total variance of the input samples estimated in the absence of interference.Time Domain Complex Signum (TDCS): the input samples are processed with the complex signum non-linearity and the output samples are obtained as
(1)$$\tilde{y}\left[n\right]=\left\{\begin{array}{cc} \frac{y[n]}{\left|y[n]\right|} & y[n]\neq 0\\ 0 & y[n]=0 \end{array}\right.$$
where y[n] are digital samples provided by the receiver front-end and *n* is the time index. A sampling frequency $f_{s}$ is assumed.Frequency Domain Pulse Blanking (FDPB) implements frequency domain excision: the input samples are at first brought into the frequency domain using a Discrete Fourier Transform (DFT)/Fast Fourier Transform (FFT) operation. Blanking is then applied and frequency samples with a magnitude greater than a threshold are set to zero. Finally, the blanked frequency domain samples are brought back in the time domain. In addition, in this case, we used a threshold equal to three times the standard deviation of the frequency domain samples estimated in the absence of interference.Frequency Domain Complex Signum (FDCS): complex signum non-linearity (Equation 1) is applied to the frequency domain samples. As for the FDPB case, DFT/FFT and inverse operations are used to transform the samples between time and frequency domains.

RIM techniques are the most computational demanding approaches among the interference defences considered in this paper. They operate on the digital samples provided by the receiver front-end, which are characterised by a rate which is several thousands of time faster than that of the correlators and of the GNSS measurements. The high data rate of the input samples makes RIM techniques significantly more computationally demanding than the PLI and T-RAIM, which operates on the correlators and the GNSS measurements.

Time domain techniques have a linear complexity with respect to the number of input samples. This implies that a fixed number of operations is required for each input sample. The complexity of frequency domain RIM techniques is dictated by the transformation used to bring the sample to and from the frequency domain. When a DFT/FFT is used, the complexity is proportional to Nlog(N), where *N* is the DFT/FFT size.

A detailed analysis of RIM techniques and of their computational requirements is out of scope of this paper and can be found in [9] and references therein.

### 2.2. Post-Correlation Defences

Interference mitigation techniques can also be implemented at the post-correlation level using signal correlators to compute quality indicators, which can be used for excluding measurements. The Carrier-to-Noise power spectral density ratio ($C/N_{0}$) and LIs are commonly used to determine if a tracking loop is correctly processing a GNSS signal [10]. Moreover, $C/N_{0}$ measurements can be combined to detect the presence of jamming [15].

In this paper, we adopted a standard PLI from the literature [10] to screen measurements. The PLI is computed as:(2)$$PLI_{k}=\frac{P_{I}^{2}-P_{Q}^{2}}{P_{I}^{2}+P_{Q}^{2}}$$
where $P_{I}$ and $P_{Q}$ are the in-phase and quadrature components of the prompt correlator. The index *k* is used to denote the different processing epochs. For the Galileo E1C signal, a prompt correlator is produced each 4 ms whereas, in the E5B case a correlator is computed each 1 ms. $PLI_{k}$ can be expressed as the cosine of twice the residual phase error and assumes values in the [−1,1] range [10]. Signal phase lock is achieved when $PLI_{k}$ is close to 1.

In order to improve the reliability of the PLI, an exponential filter has been adopted to reduce the impact of noise. In particular, a filtered PLI has been obtained as
(3)$${\bar{PLI}}_{k}=\alpha {\bar{PLI}}_{k}+\left(1-\alpha \right)PLI_{k}$$
where α∈[0,1) is the filter forgetting factor. In this work, α=0.99. If ${\bar{PLI}}_{k}<T_{PLI}$, phase lock is declared lost, the associated signal is no longer tracked and no measurement is generated. In this way, the PLI operates as a first mechanism preventing the generation of faulty measurements. In this work, $T_{PLI}$, the PLI decision threshold, was set to 0.2. This value was found empirically as a good compromise between the ability to detect loss of lock conditions and erroneously excluding potentially good measurements.

### 2.3. Navigation Solution Defences

In this section, the defence layer implemented at the navigation solution level is described. A Galileo-only timing solution is considered. Since it is computed considering the receiver position known, the timing solution is obtained as the weighted mean of the available measurements. Considering the mean as the solution of a least square problem, it can be written as:(4)$$x={\left(h^{T}\cdot W\cdot h\right)}^{-1}\cdot h\cdot W\cdot z$$
where *x* is the local clock bias estimate with respect to Galileo System Time (GST), h is a column vector of ones with length equal to the number of measurements, *W* is the weighting matrix which is diagonal and whose elements are function of the $C/N_{0}$ [16]; *z* is the vector containing the measurements corrected for all the error terms and for the user-to-satellite distance, which is obtained from the known user position.

Since *W* is diagonal, w=h·W is the column vector containing the weights on the diagonal of *W* and $\left(h^{T}\cdot W\cdot h\right)=\sum_{i=0}^{N_{m}-1}w_{i}$ where $N_{m}$ is the number of measurements and $w_{i}$ are the different weights. In this way, (Equation 4) becomes
(5)$$x=\frac{1}{\sum_{i=0}^{N_{m}-1}w_{i}}\sum_{j=0}^{N_{m}-1}w_{j}z_{j}$$
where ${\left\{z_{j}\right\}}_{j=0}^{N_{m}-1}$ is the measurement set.

The residuals, r, are defined as:(6)r=z−h·x
and represent the discrepancies between the estimated solution and the measurements.

The T-RAIM algorithm implemented in this work exploits residuals: its schematic representation is shown in Figure 2. The algorithm is derived from the Forward–Backward scheme presented in [16,17]. The technique includes two phases: Forward (light blue box) and Backward (orange box). In the first part, four different tests are used: the TPL test that verifies if the geometry is robust enough for supporting integrity checks; then a Global Test (GT) is used to verify the consistency of the whole set of measurements; if the GT fails, possible outliers are flagged by a Local Test (LT). If one or more outliers are identified, a separability test is carried out to verify if the outliers are not too much correlated with other measurements. The Forward phase is iteratively repeated until no outliers are identified or the solution is declared unreliable. The second part of the algorithm is only based on the GT: in case of multiple exclusions, the measurements are singularly re-introduced and the solution is re-computed and declared reliable if the GT is passed. This second part of the algorithm prevents wrong exclusions due to the order of the rejections.

The different tests used for T-RAIM are briefly discussed in the following sections.

#### 2.3.1. TPL Check

The robustness of the geometry is assessed using the TPL which is compared to the Timing Alarm Level (TAL) set to 30 ns for this study.

The TPL is composed by two terms, the first related to measurement bias (indicated as Weighted Approximated Time Protected (WATP)) and a second one related to the measurements noise $HPL_{n}$. To compute these two elements, the following procedure is adopted. First of all, the Weighted Timing Slope (WTS) is computed as:(7)$$WTS_{i}=\frac{a_{i}}{S_{i,i}}$$
where $a_{i}$ is the *i*th element of the row vector:(8)$$a={\left(h^{T}\cdot W\cdot h\right)}^{-1}\cdot h^{T}\cdot W$$
and $S_{i,i}$ is the *i*th diagonal element of the matrix
(9)S=I−h·a
where *I* the identity matrix of size $N_{m}\times N_{m}$. Note that the WTS is computed for all the measurements, that is for *i* in $\left[0,N_{m}-1\right]$. Moreover, a is a left pseudo-inverse of h and the coefficients,
(10)$$a_{i}=\frac{w_{i}}{\sum_{j=0}^{N_{m}-1}w_{j}}$$
weight the different measurements in (Equation 5). The WTS for the *i*th measurement is obtained by further normalising $a_{i}$ by the corresponding diagonal element of *S*.

Finally, using the maximum WTS, the WATP is computed as [18]:(11)$$WATP=\max_{i}\left(WTS_{i}\right)\cdot p_{bias}$$
where $p_{bias}$ is the the square root of the GT threshold [19].

The noise dependent part of the TPL is computed as
(12)$$TPL_{n}=K\cdot \sqrt{\sigma_{Sol}^{2}}$$
where *K* is the protection coefficient depending on the missed detection probability [20], and $\sigma_{Sol}^{2}$ is the variance of the solution.

Finally the composite TPL is obtained as:(13)$$TPL=WATP+TPL_{n}.$$

The computed value is compared with respect to TAL; if the TPL is higher than the threshold the solution is declared unreliable because the geometry is not robust enough to support integrity check.

#### 2.3.2. Global Test Check

If the TPL check is passed, a consistency check on the whole set of measurements is performed; this test is commonly known as GT. The test is based on residuals and the test variable is a weighted quadratic form of the residuals [21,22,23]:(14)$$D_{GT}=r^{T}\cdot W\cdot r.$$
$D_{GT}$ is compared with respect to the threshold $T_{GT}$:(15)$$T_{GT}=\chi_{1-P_{FA},\left(N_{m}-1\right)}^{2}$$
where $P_{FA}$ is the probability of false alarm (set to 0.1%), and $N_{m}$ is the number of measurements.

If $D_{GT}$ is lower than $T_{GT}$, the solution is declared reliable and no additional tests are performed, otherwise an LT is carried out.

#### 2.3.3. Local Test Check

The LT is performed to flag possible outliers and its detection variable is the normalised residual computed as:(16)$$D_{LT}=\left|\frac{r_{i}}{\sqrt{C_{i,i}}}\right|$$
where $r_{i}$ is the residual of the *i*th satellite (with *i* from 0 to $N_{m}-1$), $C_{i,i}$ is the *i*th element of the diagonal of the residual covariance matrix.

The threshold of the local test, $T_{LT}$, is equal to the abscissa corresponding to the probability value $1-P_{FA}/2$ of a normal distribution. If more than one normalised residual exceeds the threshold, the largest one is regarded as a possible blunder. If all the normalised residuals are below the threshold, the solution is declared unreliable due to an inconsistency between the results of the GT and LT checks.

#### 2.3.4. Separability Check

Before excluding the possible outliers pointed out by the LT, the correlation among the measurements is assessed. In this case, the correlation coefficient is used as detection variable and is computed as [24]:(17)$$D_{SC}=\frac{C_{i,j}}{\sqrt{C_{i,i}\cdot C_{j,j}}}$$

If $D_{SC}$ is lower than the separability threshold (0.8 in this study), the identified outlier is excluded and the Forward phase is repeated until no more outliers are present or the solution is declared unreliable. If $D_{SC}$ is higher than the threshold, the solution is declared unreliable since a wrong exclusion could be due to the effect of large outliers.

## 3. Data and Setup

A dedicated experimental setup has been designed and implemented to assess the interaction among the different defence layers. The setup is the same one used in [9] and thus is only briefly described here. A schematic representation of the setup is shown in Figure 3.

Clean GNSS signals were collected using a geodetic antenna placed on the rooftop of an office building in the Joint Research Centre (JRC) campus in Ispra. The antenna was in open-sky conditions and its position was pre-surveyed. In order to contaminate clean signals, a jammer was used. The jammer was placed inside a shielding box whose output was connected to a variable power attenuator used to control the jammer power. Then, the clean GNSS signals and interference signal were mixed using a signal combiner. The output of the signal combiner was connected to a Universal Software Radio Peripheral (USRP) used to collect In-phase Quadrature (I/Q) samples that were stored on disk. Using a customised Matlab software receiver the samples were processed several times, applying the techniques described in Section 2. The interested reader can find additional details about the setup in [9]. Two different tests are considered in this work: Test 1 with GPS L1 and Galileo E1 signals and Test E5B where only Galileo E5B signals were present. In both tests, the same jammer was used. We adopted the same nomenclature of [9] in order to allow a direct comparison of the results.

## 4. Results

The first layer of defence considered in this paper is represented by RIM techniques at the pre-correlation level, hence five baseline configurations (without any additional defence) can be considered including four RIM algorithms and the case without mitigation. Then for each of these five baseline configurations, three cases are considered:PLI on and T-RAIM off (PLI);PLI off and T-RAIM on (T-RAIM);PLI and T-RAIM both on (PLI + T-RAIM).

The cascade effects of the mitigation techniques are assessed considering:number of satellites used in the timing solution;TPL, in particular the variation of the TPL is considered;maximum residual;frequency stability, evaluated using the overlapping Allan Deviation (ADEV) [25,26].

### 4.1. Test 1 Results

In this section, the results obtained considering Test 1 are presented. In order to avoid the repetition of a similar finding, only results relative to Galileo E1 are presented. Results related to GPS L1 Coarse Acquisition (C/A) signals can be found in [13].

The first parameter assessed is the number of satellites used in the timing solution; in order to evaluate the impact of the different defence layers for each configuration, the variation of the number of satellites with respect to the baseline configuration is shown in Figure 4.

In all tests considered, an increasing jamming power was progressively injected [9]. During the first 300 s of Test 1, the jamming power can be considered negligible. For this reason, no differences were observed before about 300 s from the start of the tests and the following plots consider epochs after 300 s. This choice was adopted to improve the readability of the plots. Between 300 and 600 s when the jammer power is progressively increased, few exclusions are performed for the “no mitigation” and FDCS cases. These exclusions are performed by the T-RAIM algorithm even if the PLI was still declaring all satellites locked. Time domain RIMs is effective in mitigating the impact of jamming and no exclusions were performed. Finally, for FDPB early exclusions/losses of lock are clearly visible with a degradation of the performance with respect to the “no mitigation” case. From the exclusion point of view, T-RAIM seems to be more reactive than the PLI; such behaviour strongly depends on the thresholds used for the two techniques.

In the final part of the test, for the configurations with PLI and T-RAIM, the exclusions are mainly performed by the PLI, which reduces the satellite availability. With a low measurement redundancy, T-RAIM is unable to perform additional exclusions.

For an easier comparison among the different configurations with both, PLI and T-RAIM activated the variation of used satellites, which is shown in Figure 5 for the different pre-correlation mitigation techniques. From the comparison, it clearly emerges that FDPB is characterised by an increased number of exclusions/losses of locks which occur earlier with respect to the other cases.

“No mitigation” and FDCS cases have similar performance, while time domain RIM reduces the number of excluded satellites leading to a reliable timing solution for almost the entire test duration.

Statistical parameters, mean and maximum value of the variation of the number of satellite used in the timing solution and for Test 1 are reported in Table 1. From the table it can be noted that the joint use of PLI and T-RAIM led to up to five excluded satellites.

The TPL has been computed for the all the configurations (including the ones without T-RAIM) and its variation with respect to the baseline configuration is shown in Figure 6. The introduction of measurement screening, either through the PLI or T-RAIM, causes an increase of TPL. This is expected since the exclusion of a measurement results in a degradation of the satellite availability and corresponding geometry. In some cases, the TPL occurs only at the very end of the test, showing the ability of pre-correlation mitigation techniques to reduce the impact of jamming.

The TPL varies according to the satellite availability: larger and earlier increases are observed for the FDPB cases, while the FDCS and “no mitigation” cases have a similar behaviour. Finally, time domain RIM techniques show an increased TPL (up to 20 ns) only in the very last part of the test.

The TPL variation is shown in Figure 7 as a function of time and for the five configurations with PLI and T-RAIM activated. The figure allows a direct comparison among the different configurations. As for the previous figures, all the configurations provide similar performance during the first 300 s of the tests. These epochs are not depicted in Figure 7 since no increase with respect to the baseline case was visible. Between 300 and 500 s, the degradation introduced by the FDPB is clearly visible. Finally, in the very last part of the test, only TDPB and TDCS were able to effectively limit the TPL increase. Mean and maximum values of the variation of the TPL are reported in Table 1. From the table, it can be noted that a mean degradation of a few nanoseconds is introduced when the mitigation techniques are used. The maximum increase of the TPL was about 25 ns for the FDCS case.

The maximum value of the residuals is shown in Figure 8 as a function of time, where the case without pre-correlation defences is considered. The configuration without RIM has been used to better isolate the impact of the two post-correlation techniques used. The figures relative to the cases when the four RIMs are activated are not presented to avoid repetition of similar finding. In the upper box, the baseline configuration is shown with (red line) and without T-RAIM (blue line): the impact of the T-RAIM algorithm is clearly visible; the red line is always lower than the blue one. Moreover, the difference is more visible as the jammer power increases, demonstrating that the T-RAIM algorithm is able to exclude measurements that are strongly affected by the jamming signal. In the central box, the impact of the activation of the PLI is analysed. In addition, in this case the red line is lower than the blue one; only in the very last part of the test are the differences more visible. Under these conditions the PLI algorithm declares loss of lock for some satellites. For the baseline configuration with and without PLI, an increased value of the residuals can be observed when the jamming power increases, while for the T-RAIM this phenomenon is not visible. In the lower box, the configurations with PLI are considered with and without T-RAIM: the joint use of the PLI and T-RAIM further reduces the residual values, showing the advantages of using multiple defence layers.

In order to analyse the impact of the pre-correlation techniques, the maximum value of the residuals as a function of time is shown in Figure 9. In the upper box, frequency domain techniques are considered while in the lower box time domain approaches are shown: all the configurations have both PLI and T-RAIM activated. For the frequency domain cases, an increase of the residual values is clearly visible for the FDPB while the FDCS and “no mitigation” cases behave similarly. The advantages of the application of time domain RIMs are evident; a sensible reduction of the maximum residual can be noted mainly when the jammer power is higher (i.e., after 300 s).

The ADEV of the timing solutions is shown in Figure 10, the configurations without pre-correlation defences are considered in the left side while on the right the configurations using RIMs with PLI and T-RAIM are provided.

For the configurations with T-RAIM activated, the ADEV curves have been computed using only reliable epochs; for the configurations without T-RAIM, the ADEVs have been estimated considering either all epochs (light blue and green lines) or considering reliable epochs only. When only reliable epochs are considered, the curves are very close to each other and only small differences can be noted for averaging time interval lowers than 3 s. Completely different curves are obtained considering all the epochs showing the impact of removing unreliable solutions.

The ADEVs curves using RIMs show that all the configurations converge to similar values for averaging time intervals larger than 5 s while, for τ smaller than 5 s, a sensible degradation with respect to the “no mitigation” case can be noted when FDPB is used. Finally, an increased stability is obtained using time domain RIMs. For all the configurations, only solutions flagged as reliable by the T-RAIM algorithm were used for the estimation of the ADEV. When RIM techniques are used, an increased reliability of the solution was observed and a larger number of epochs was available for the evaluation of the ADEV. For this reason, ADEV curves were evaluated considering larger averaging times in the case of time domain RIM.

### 4.2. Test E5B Results

In this section, the results of Test E5B using Galileo E5Bsignals are presented.

In Figure 11, the variation of the number of used satellites as a function of the epoch is shown. In the figure, the five cases considering the different pre-correlation mitigation techniques are shown in different boxes. The largest variations (up to two exclusions) are observed for the “no mitigation” and FDPB cases while for the other approaches a difference of maximum one satellite can be noted for most of the duration of the test. For the “no mitigation” case, a larger number of exclusions is performed by the LI and T-RAIM algorithms, while when a pre-correlation mitigation technique is used the number of exclusions is reduced.

In order to better compare the performance of the different techniques, the five cases are jointly analysed in Figure 12; in the figure, all the configurations have PLI and T-RAIM activated. In this case, all the mitigation techniques provide some margin with respect to the “no mitigation” case that is unable to provide reliable solutions after 445 s from the start of the test. In this respect, Figure 12 also reports the epoch index of the last reliable epoch obtained for each configuration. In addition, in this case, time domain RIM achieves the best performance and TDPB is able to provide a reliable solution almost till the end of the test.

Mean and maximum values of the variation of the number of satellite used in the timing solution and for Test E5B are reported in Table 2. From the table, it emerges that the joint use of PLI and T-RAIM led to up to two exclusions: the limited number of exclusions for this test is essentially due to the low number of visible satellites.

In Figure 13, the variation of the TPL with respect to the baseline configurations is shown: from the figure, the impact of the increasing jammer power on the TPL is clearly visible. When no mitigation is applied, an increase of about 20 ns can be appreciated after about 430 s from the start of the test. These values were observed when using time domain mitigation only toward the end of the test. The larger variations are relative to the PLI+T-RAIM case, and it is due to the increased number of excluded satellites.

The configurations with PLI and T-RAIM are compared in Figure 14: the increase of TPL clearly emerges from the figure. The most affected case is the one without mitigation (blue line): measurements are severely affected by jamming and thus excluded by PLI and T-RAIM with a consequent degradation of the TPL. In this test, all mitigation techniques are able to limit the increase of TPL.

Mean and maximum values of the variation of the TPL are reported in Table 2. From the table, it can be noted that the mean degradation of the TPL is almost zero when the mitigation techniques are used. The maximum value reaches about 20 ns for the TDCS case.

The maximum residual value is shown in Figure 15 as a function of time for the configurations without pre-correlation techniques. From the figure, a reduction of the maximum residual when using PLI and T-RAIM is evident (upper and central box). For the lower box, it can be noted that the joint application of PLI and T-RAIM led to a further reduction of the maximum residual values; this reduction is more evident toward the end of the test when the jammer power is higher.

The impact of the different pre-correlation mitigation techniques on the maximum residual value is analysed in Figure 16. In the figure, all the configurations have PLI and T-RAIM activated. In the upper box, frequency domain techniques are compared with respect to the “no mitigation” case, while in the lower box time domain techniques are considered. For all the configurations, no increase is visible in terms of maximum residual. In particular, the time domain techniques show limited residual values even in the presence of strong interference. This is mainly due to the use of PLI and T-RAIM.

The impact on the stability of the timing solution is analysed using the ADEV; the curves for the configuration without RIM techniques are shown in the left part of Figure 17. Only small differences at very short term can be noted between the different cases. Note that the ADEV has been computed using the first 450 epochs of the tests before observing significant drifts in the timing solutions. This limits the variations to short averaging times.

In the right part of Figure 17, the ADEV curves for the configurations with different RIM techniques and with PLI and T-RAIM activated are shown: only reliable epochs are used for the estimation of the ADEV. In this case, all the curves overlap with minimal differences in the short term. In particular, a small degradation can be observed for the FDPB case.

## 5. Conclusions

In this paper, an experimental evaluation of the performance and interactions of three levels of receiver defences against jamming and interference has been performed. The work focused on RIM, PLI and T-RAIM. While the first layer of defence at pre-correlation level, e.g., RIM, significantly improves receiver performance, a further screening of the measurements is required to obtain reliable timing solutions. PLI and T-RAIM complete the work performed by RIM leading to reliable timing solutions.

The performance has been evaluated using real GNSS signals mixed with an increasing level of interference: the tests presented are performed on two frequencies (E1 and E5B) to demonstrate that the proposed approach is suitable for different frequencies and modulations.

From the analysis it emerged that a proper mitigation technique at pre-correlation level improves the reliability of the measurements leading to a reduced number of exclusions performed by the FDE algorithm. For the type of jammers used in the tests, time domain RIM is the most effective approach leading to a reliable timing solution for almost the entire duration of the test. When the interference level is very high, the exclusions are mainly performed by the PLI which reduces the satellites availability. With a low measurement redundancy, T-RAIM is unable to perform additional exclusions.

In a multi-layers approach the proper selection of the mitigation techniques is fundamental to avoid performance degradation in the presence of jamming.

## Figures and Tables

**Figure 1 sensors-21-07787-f001:**
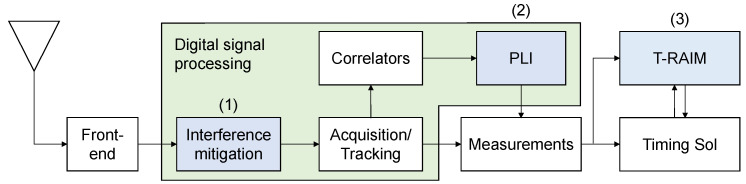
Schematic representation of the different processing blocks implemented for the evaluation of the timing solution: from the antenna to the final solution. Interference defence blocks are indicated in light azure.

**Figure 2 sensors-21-07787-f002:**
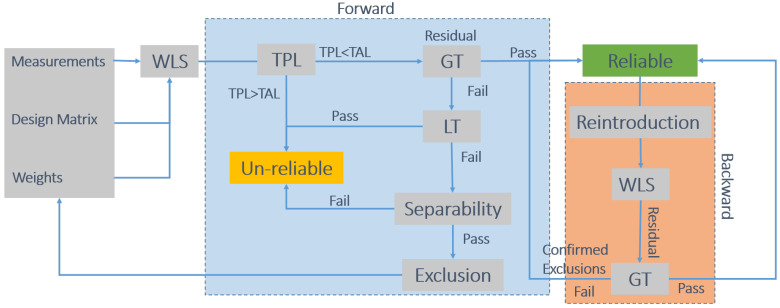
Schematic representation of the T-RAIM algorithm.

**Figure 3 sensors-21-07787-f003:**
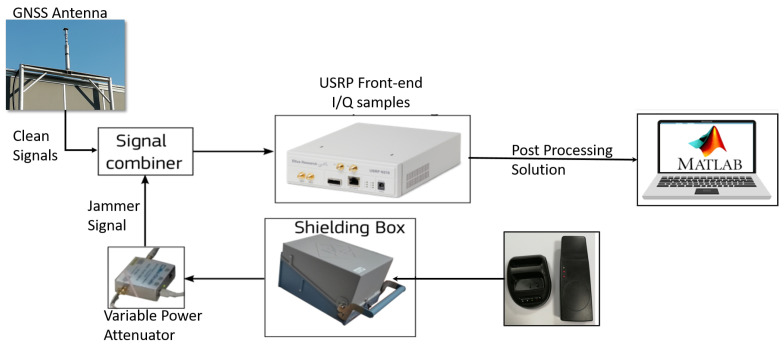
Schematic representation of the setup developed for the experimental evaluation of the interaction among the defence layers.

**Figure 4 sensors-21-07787-f004:**
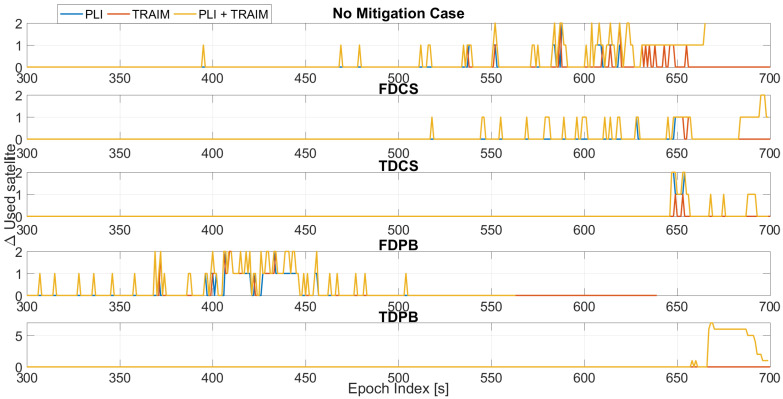
Variation of the number of used satellite with respect to to the baseline cases without PLI and T-RAIM. Test 1.

**Figure 5 sensors-21-07787-f005:**
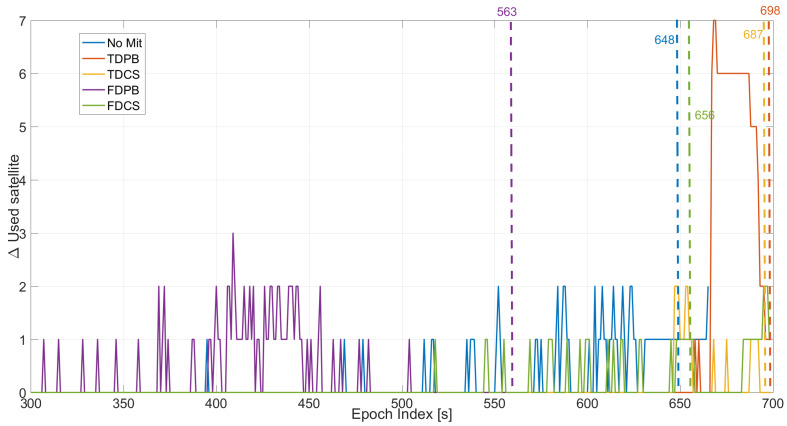
Variation of the number of satellites used in the timing solution, considering different mitigation strategies. All the configurations use PLI and T-RAIM. Test 1.

**Figure 6 sensors-21-07787-f006:**
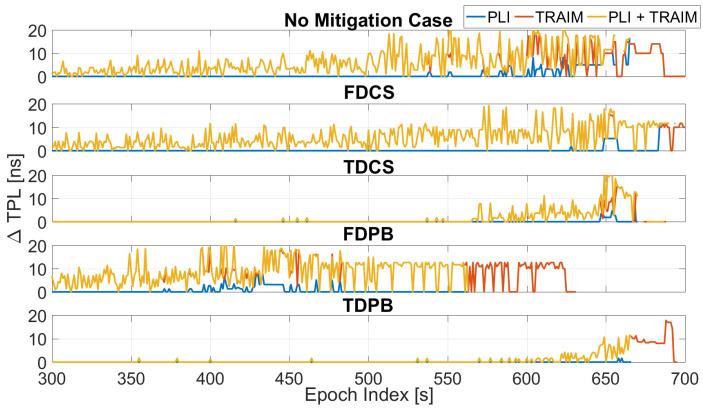
Variation of TPL computed for the different configurations with and without PLI and T-RAIM. Test 1.

**Figure 7 sensors-21-07787-f007:**
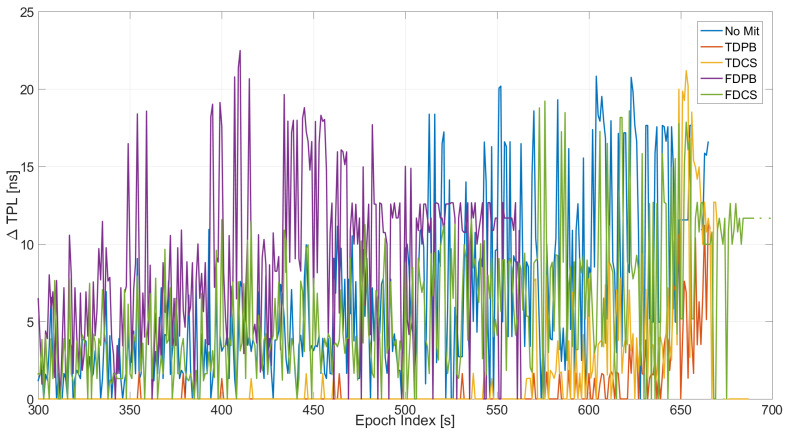
TPL for the different mitigation strategies. All the configurations have PLI and T-RAIM activated. Test 1.

**Figure 8 sensors-21-07787-f008:**
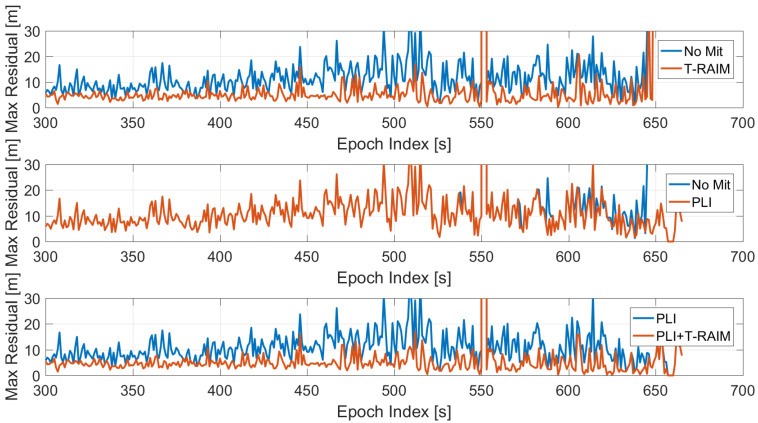
Maximum value of the residuals for the standard case without pre-correlation interference mitigation. Test 1.

**Figure 9 sensors-21-07787-f009:**
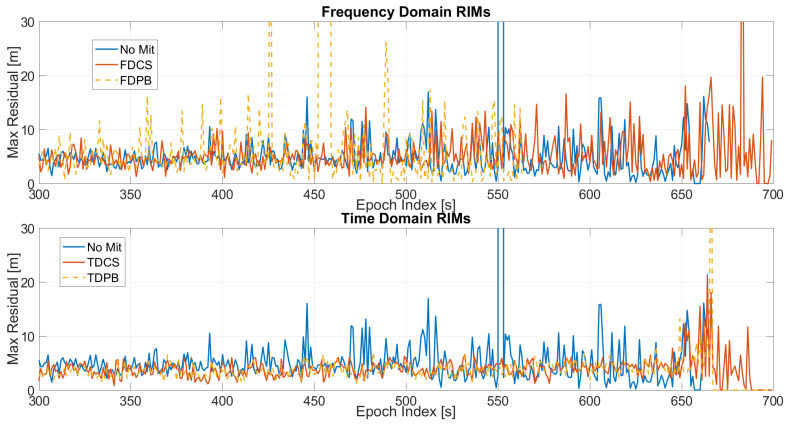
Maximum residual values for the different mitigation techniques. All the configurations have PLI and T-RAIM activated. Test 1.

**Figure 10 sensors-21-07787-f010:**
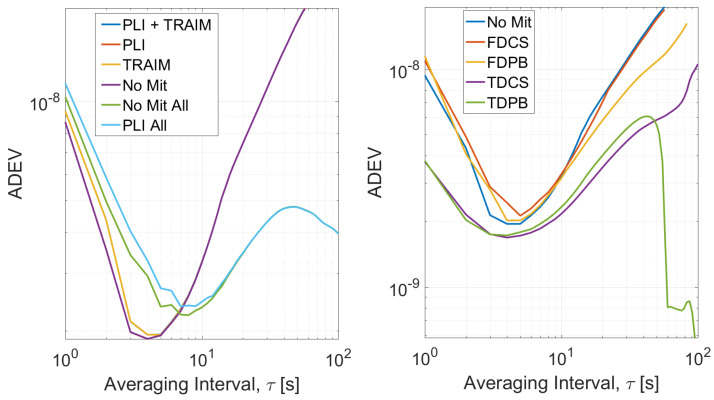
ADEV curves obtained for Test 1. (**Left box**) Processing without pre-correlation interference mitigation. (**Right box**) Processing with pre-correlation interference defence and PLI and T-RAIM.

**Figure 11 sensors-21-07787-f011:**
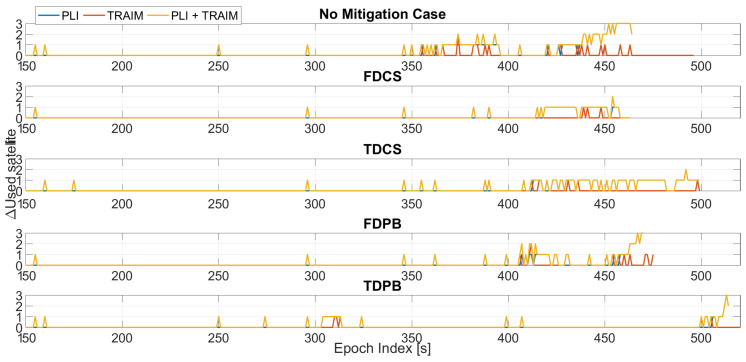
Variation of the number of used satellite with respect to the baseline cases without PLI and T-RAIM. Test E5B.

**Figure 12 sensors-21-07787-f012:**
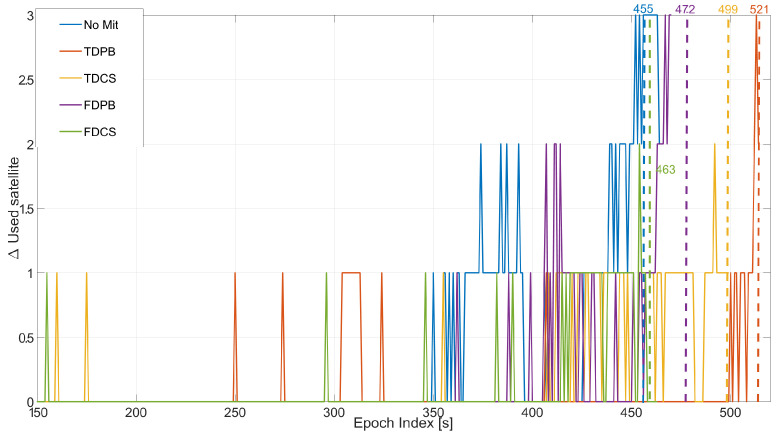
Variation of the number of satellites used in the timing solution, considering different mitigation strategies. All the configurations used PLI and T-RAIM. Test E5B.

**Figure 13 sensors-21-07787-f013:**
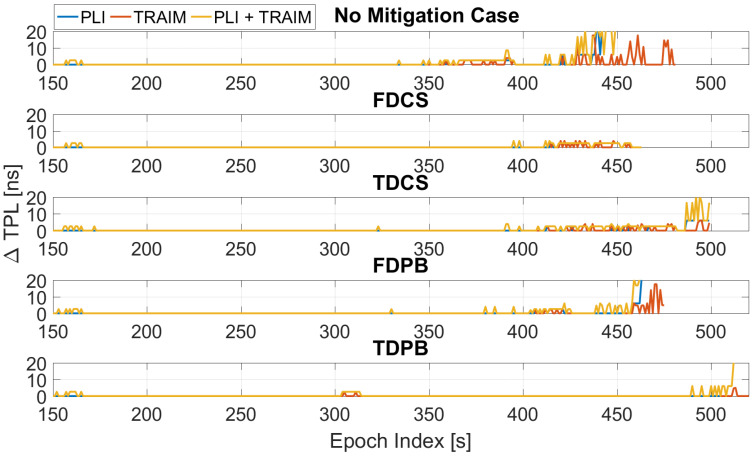
Variation of TPL computed for the different configurations with and without PLI and T-RAIM. Test E5B.

**Figure 14 sensors-21-07787-f014:**
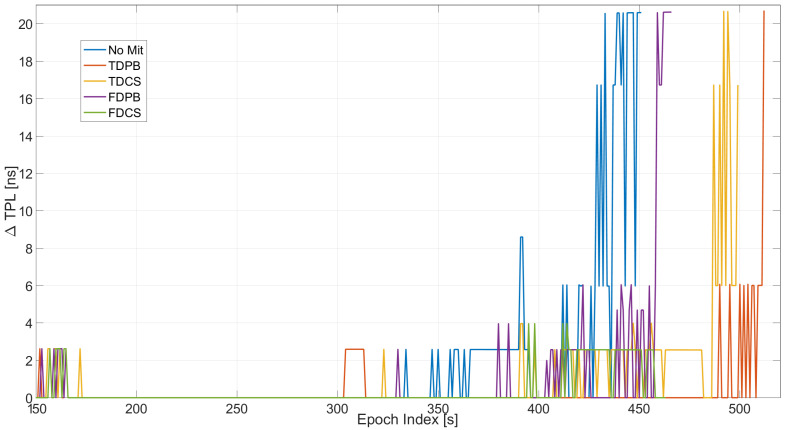
Variations of TPL for the different mitigation strategies. All the configurations have PLI and T-RAIM activated. Test E5B.

**Figure 15 sensors-21-07787-f015:**
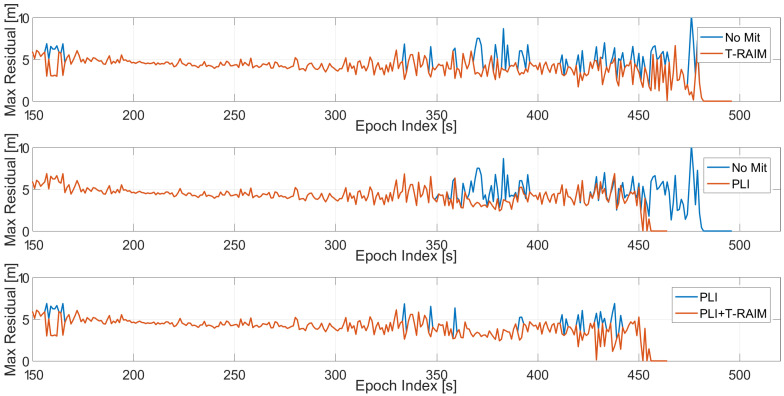
Maximum value of the residual for the standard case without pre-correlation interference mitigation. Test E5B.

**Figure 16 sensors-21-07787-f016:**
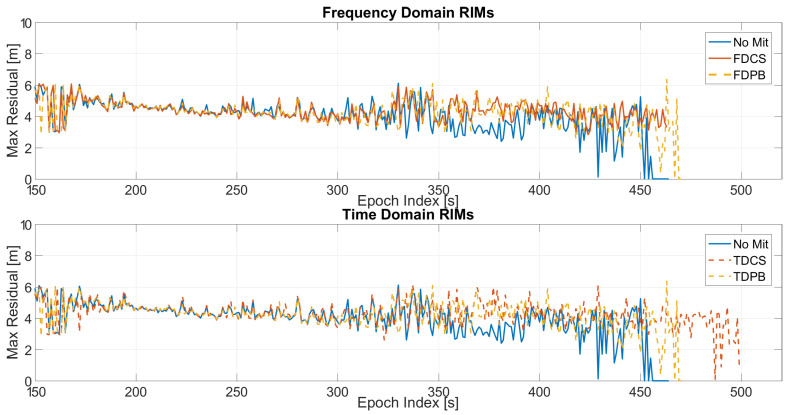
Maximum residual values for the different mitigation techniques. All the configurations have PLI and T-RAIM activated. Test E5B.

**Figure 17 sensors-21-07787-f017:**
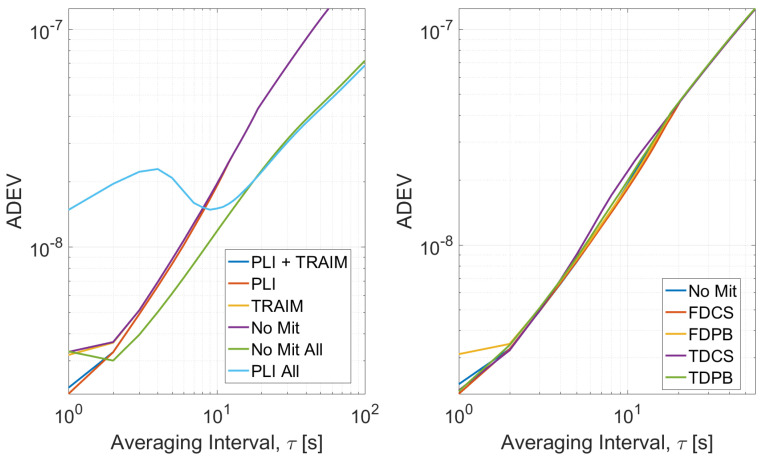
ADEV curves obtained for Test E5B. (**Left box**) Processing without pre-correlation interference mitigation. (**Right box**) Processing with pre-correlation interference defence and PLI and T-RAIM.

**Table 1 sensors-21-07787-t001:** Statistical parameters of the variation of the number of satellite used in the timing solution and variation of the TPL for the different configurations. Test 1.

Configuration	Variation Sat Used		Variation TPL (ns)	
	Mean	Max	Mean	Max
FDCS + T-RAIM	0.11	1	3.45	19.22
FDCS + PLI + T-RAIM	0.13	5	3.50	22.11
TDCS + T-RAIM	0.40	2	0.73	15.39
TDCS + PLI + T-RAIM	0.44	5	0.91	21.01
FDPB + T-RAIM	0.92	4	4.53	19.02
FDPB + PLI + T-RAIM	1.01	5	4.70	25.51
TDPB + T-RAIM	0.41	1	0.42	17.76
TDPB + PLI + T-RAIM	0.66	5	0.73	21.41
No Mit + T-RAIM	0.04	2	3.4	18.59
No Mit + PLI + T-RAIM	0.14	5	3.6	23.21

**Table 2 sensors-21-07787-t002:** Statistical parameters of the variation of the number of satellite used in the timing solution and variation of the TPL for the different configurations. Test E5B.

Configuration	Variation Sat Used		Variation TPL (ns)	
	Mean	Max	Mean	Max
FDCS + T-RAIM	0.03	1	0.20	3.98
FDCS + PLI + T-RAIM	0.10	1	0.28	3.96
TDCS + T-RAIM	0.16	1	0.27	6.06
TDCS + PLI + T-RAIM	0.28	1	0.71	20.66
FDPB + T-RAIM	0.04	2	0.43	17.53
FDPB + PLI + T-RAIM	0.05	2	0.67	20.62
TDPB + T-RAIM	0.05	2	0.13	6.07
TDPB + PLI + T-RAIM	0.06	2	0.26	20.70
No Mit + T-RAIM	0.06	2	0.61	17.62
No Mit + PLI + T-RAIM	0.22	3	1.24	20.60

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
