# Peer review of "Multi-Layer Defences for Robust GNSS Timing Retrieval†"

_sensors, 2021, doi:10.3390/s21237787_

Round 1

Reviewer 1 Report

Thank you for your interesting contribution. I appreciate the huge amount of work that has been conducted to achieve the results. The concepts, metrics, and definitions are remarkable. However, the manuscript requires some modifications, and there are points to be clarified:

  1. The structure of the manuscript is unbalanced. Section 2 is defined in 5 pages, while Section 3 is only a paragraph! The same comparison can be conducted with Sections 4 and 3. I absolutely understand that you did not want to rewrite the setups which are already published in other works. However, adding different photos of the actual setup and more explanations can improve the understandability of the research. They also balance the manuscript. 
  2. I cannot understand the reason and evidence that Fig. 4 provides. Any prove from this figure can be concluded from Fig. 3. Massive information is combined in one figure, while Fig. 3 expresses them in a better way. The same issue exists in Fig. 6, Fig. 11, and Fig. 13. Maybe I missed something in the context about the necessity of these figures. Therefore, I would appreciate explanations.
  3. Why only three cases are selected in figures 7 and 14? Are there any noteworthy points in these cases in comparison to others?
  4. A quantitative comparison between methods and different cases can improve the reader's vision of the results. I suggest adding a table of each parameter's mean or any metric that you consider suitable.

Author Response

The authors wish to thank the editor and the reviewers for the thorough and useful evaluation of their paper. The authors have benefited from the reviewer/editor comments and insights and have revised the paper according to their suggestions.

Detailed replies to the reviewers’ comments are provided in the attached file.

Reviewer 2 Report

1.In Section 3, Data and Setup, the experimental setup mentioned can be briefly introduced, and it is best to explain it with figures.

2.The comparison of various algorithms in this paper is innovative, but not advanced enough.

3.In Section 4, it is worth considering adding the comparison of the calculation time among them.

4. Part of the content is repeated throughout the paper.

5.It is hoped that author pay attention to the logical relationship between the various parts of the article to ensure that the structure of the article is compact.

Author Response

(The authors gave the same response as above.)

Reviewer 3 Report

The article is well conceived. The results are well and clearly presented. The conclusion is also clear and correctly written. My only objection is that the paper was not written according to the rules of the journal (see instructions for authors). Missing Back Matter, I think this part of the article is very important in cases like this (reports on conflict conflicting).

Author Response

The authors wish to thank the editor and the reviewers for the thorough and useful evaluation of their paper. The authors have benefited from the reviewer/editor comments and insights and have revised the paper according to their suggestions.

Detailed replies to the reviewers’ comments are provided in the Attached file.

Round 2

Reviewer 1 Report

The overall quality of the manuscript is acceptable.